# Getting psychiatry on the move—Implementation and evaluation of Braining, a structured physical exercise intervention in outpatient psychiatry: A convergent-parallel mixed methods study

Rebecka Mac [1,2*], Erika A. Saliba-Gustafsson[1,2], Åsa Anger[1,2], Carl Johan Sundberg[3,4], Lina Martinsson[1,2], Anna Bergström[3,5,6‡], Sigrid Salomonsson[1,2‡]

1 Centre for Psychiatry Research, Department of Clinical Neuroscience, Karolinska Institutet, Stockholm, Sweden, 2 Stockholm Health Care Services, Region Stockholm, Stockholm, Sweden, 3 Department of Learning, Informatics, Management and Ethics, Karolinska Institutet, Stockholm, Sweden, 4 Department of Physiology and Pharmacology, Karolinska Institutet, Stockholm, Sweden, 5 Department of Women's and Children's Health, Uppsala University, Uppsala, Sweden, 6 Centre for Epidemiology and Community Medicine, Region Stockholm, Stockholm, Sweden

‡ shared last authorship on this work.
* rebecka.mac@ki.se

## Abstract

### Background

Physical exercise can improve mental health outcomes, yet patients with mental illness often require structured support. Braining is a 12-week structured physical exercise intervention developed within Swedish outpatient psychiatry as a clinical package. Although preliminary unpublished findings suggest positive patient outcomes, its implementation within psychiatric services has not previously been evaluated. This study aimed to explore and explain healthcare workers' perceptions of the acceptability, appropriateness, and feasibility of Braining in an outpatient psychiatric setting to inform future implementation efforts.

### Methods

A convergent parallel mixed-methods design was used with healthcare workers from outpatient psychiatry units specialising in substance use disorders. Quantitative data from the Acceptability of Intervention Measure, Intervention Appropriateness Measure, and Feasibility of Intervention Measure were collected at 1, 4, and 12 months and analysed descriptively. Qualitative data from three focus group discussions with eight Braining team members at 4 and 12 months were analysed abductively using qualitative content analysis.

**Data availability statement:** The data underlying this study contain sensitive information from healthcare staff and cannot be shared publicly due to legal and ethical restrictions imposed by the Swedish Ethical Review Authority and the data controllers (Karolinska Institutet and Stockholm Healthcare Services, Region Stockholm). Data are available for researchers who meet the criteria for access to confidential data. Requests should be directed to Karolinska Institute Research Data Office (rdo@ki.se), and will require approval by the Swedish Ethical Review Authority and the data controllers.

**Funding:** This work was supported by:
• The regional agreement on medical training and clinical research (ALF Medicin), project "Braining – implementation and scientific evaluation of structured physical activity for patients together with staff in psychiatric care" (FoUI-999932). • The regional agreement on medical training and clinical research (ALF Medicin), project "Braining – structured exercise treatment for staff and patients jointly implemented in Region Stockholm. Multicenter study: RCT, effects on physical and mental health, functioning, health economics, biobank, interview" (FoUI-987442). • The Alcohol Research Council of Systembolaget (Systembolagets Alkoholforskningsråd), project "Braining – physical exercise for improved health and reduced alcohol craving in addiction care" (FO2023-0058). • Forte (Swedish Research Council for Health, Working Life and Welfare), planning grant for "Braining – implementing physical activity for patients and personnel in psychiatry" (Dnr 2024-01863). • Author RM was partly supported by the Krica Foundation to carry out this work. The funders had no role in study design, data collection and analysis, decision to publish, or preparation of the manuscript.

**Competing interests:** The authors have declared that no competing interests exist.

## Results

Acceptability, appropriateness, and feasibility ratings remained favourable over 12 months (≥16/20). Qualitative findings reflected participants' positive views on Braining's relevance for patients, its fit with clinical workflows, and its support for interprofessional collaboration. Participants perceived improvements in patients' quality of life, benefits of joint staff–patient exercise sessions, and enhanced social connectedness. Reported barriers included variable engagement among healthcare workers outside the Braining team and challenges in reaching younger patients.

## Conclusions

Braining was perceived by healthcare workers as acceptable, appropriate, and feasible in outpatient psychiatry, with sustained implementation outcomes linked to its adaptable design, team delivery, and organisational support. As a formative evaluation, the findings reflect perceived relevance rather than effectiveness, highlighting the need for strategies to enhance equitable reach and sustainability.

## Trial registration number

NCT05111756.

## Background

Mental illness, including depression, anxiety, substance use, bipolar and psychotic disorders, affects around 15% of the global population [1–3]. Although pharmacological and psychological treatments are first-line options, they have notable limitations such as side effects, adherence issues, and limited access [4–7]. There is growing evidence that physical exercise (PE) is an effective adjunctive treatment for mental illness [8], with effects comparable to antidepressants and psychotherapy [2,9]. Benefits have also been reported in substance use disorders (SUD), where high rates of comorbid depression and anxiety suggest particular relevance [10,11]. However, evidence for effects on substance use outcomes such as consumption, abstinence, and relapse remains mixed [12–14]. PE also plays an important role in improving physical health outcomes for individuals with mental illness and SUD, who face elevated risks of cardiovascular and metabolic diseases and reduced life expectancy due to factors such as sedentary lifestyle, poor nutrition, and physiological damage caused by substance use [15]. Regular PE can help reduce risk markers, improve physical fitness, and counteract the elevated morbidity and mortality seen in this group [1,11,16]. Integrating structured PE into SUD treatment programs holds significant promise for addressing these physical health disparities and potentially extending life expectancy.

Health authorities recommend counselling to promote PE, particularly in high-risk groups. Still, patients with mental illness often face barriers such as symptoms, poor physical health, fatigue and lack of support [16–18]. Although patients with mental illness value PE for its physical and psychological benefits, most do not

meet recommended activity levels independently and therefore require additional support [18]. As brief counselling is often insufficient, structured supervised exercise programmes are needed [19]. To address this gap, *Braining*—a structured PE intervention—was developed by psychiatric staff in Stockholm, Sweden. Trained clinicians deliver this intervention and integrates PE into patients' treatment plans to support them in initiating and maintaining regular exercise. A retrospective study in outpatient psychiatry demonstrated safety and good uptake of Braining [20]. Building on this, a subsequent feasibility study at the same SUD outpatient unit as the present study indicated preliminary benefits and high acceptability from the patient perspective (Anger et al., manuscript submitted). However, Braining has not yet been systematically evaluated from the perspective of healthcare workers (HCW), whose engagement is critical for successful integration into routine care. The present study, therefore, explores and explains HCWs' views on early key implementation outcomes.

Integrating interventions into healthcare is challenging [21,22]. PE illustrates this well: despite strong evidence for its benefits across mental illness [8], there is a lack of studies evaluating how PE can be systematically implemented in psychiatric care. This gap is particularly evident in SUD, where translation into clinical practice has been limited and implementation barriers remain underexplored [23]. Structured frameworks such as PRACTIS [24] highlight the need for systematic approaches, but they have rarely been applied in psychiatric or SUD contexts. Assessing implementation outcomes can help unpack some of the complexities of integrating interventions into practice by understanding and subsequently targeting aspects of the intervention and its implementation that may affect implementation success within a specific context [25]. In Proctor et al.'s Implementation Outcomes Framework (IOF), acceptability, appropriateness, and feasibility are described as "early" outcomes that precede—and predict—later measures such as fidelity, penetration, and sustainability [25]. Demonstrating favourable early implementation outcomes helps build the evidence base needed for sustainable implementation and reduces the risk that an intervention falters once it is integrated into routine practice, thereby motivating the case for scale-up. We therefore sought to explore and explain HCWs' acceptability, appropriateness, and feasibility of Braining in an outpatient psychiatric setting to inform future implementation efforts.

## Methods

This study is part of a series of clinical intervention studies on Braining. A full description of Braining and a retrospective evaluation of its clinical use have been previously published [20]. Table 1 provides a brief overview of Braining's seven core components.

### Study setting and local implementation

This study was conducted at two outpatient psychiatry units (A and B) at the Stockholm Centre for Dependency Disorders between 2022 and 2023. Both units treated adults with SUD and other psychiatric comorbidities, with Unit B focusing on young adults (18–25 years). Braining was implemented in both units from September 2021 to June 2022, following Fixsen et al.'s implementation phases [27]. Briefly, all HCWs (n = 29) were introduced to Braining, after which a Braining team was formed, consisting of a manager and four HCWs from each unit (10 team members total). The Braining team received education, training, and weekly implementation support from the research team. Two lead coordinators received 20% of full-time support from the research team. Braining was launched for patients in February 2022, with recruitment from February to March 2022. Recruited patients participated in the 12-week Braining intervention until June 2022, after which the units delivered Braining independently, with optional implementation support twice a year.

### Study design and primary implementation outcomes

This was a convergent-parallel mixed-methods study (quan + QUAL) focusing on three IOF outcomes—acceptability, appropriateness, and feasibility [25] —as outlined in Table 2. Recruitment took place from December 1st 2021, to January 31st 2023.

**Table 1. Brief overview and description of Braining's seven core components.**

| | Core components | Description of core components |
|---|---|---|
| 1 | Startup, follow-up and completion | A 12-week treatment with follow-up visits at 4, 8 and 12 weeks. Visits include somatic examination, individual goal setting and surveys. At startup, patients are screened for physical health risks and inclusion criteria. |
| 2 | The Braining team | Consists of a multidisciplinary team (including nurses, administrators, physicians, physiotherapists, and their manager) of at least 5 members. The team organises Braining through structured weekly meetings. The team also includes a lead coordinator responsible for local implementation and startup coordinators responsible for conducting core component one. |
| 3 | Braining instructor and Braining host | Healthcare workers trained to lead the Braining sessions (Braining instructor) and support patients during the class (Braining host). |
| 4 | Braining sessions | Aerobic-focused group training lasting 30–45 minutes, offered three times per week at the unit, based on the BrainingBox and including a variety of body-weight exercises [26]. Sessions are performed in close proximity to the outpatient clinic, for example indoors in an office room of approximately 20 m² or outdoors. |
| 5 | Individual check-in | Prior to each training session, patients' mental health and physical condition are briefly assessed via phone or on site. Patients also receive supportive feedback by the Braining instructor or host. |
| 6 | Booking | Patients book and cancel workout sessions independently. Participation is provided at no additional cost. |
| 7 | Motivational work | Braining is infused with motivational work, embedded in all patient interactions, including individualised goal setting. |

**Table 2. Study-specific definitions of Proctor's IOF and associated data sources.**

| Proctor et al.'s IOF [25] | Study-specific definitions | Quantitative data | Qualitative data |
|---|---|---|---|
| Acceptability | Degree to which healthcare workers believe Braining is an acceptable and satisfactory intervention. | Survey: AIM | Semi-structured FGDs |
| Appropriateness | Healthcare workers' perceived fit, relevance, or compatibility of Braining for patients with SUDs and within outpatient psychiatry. | Survey: IAM | Semi-structured FGDs |
| Feasibility | Degree to which healthcare workers believe Braining can be successfully implemented in outpatient psychiatry. | Survey: FIM | Semi-structured FGDs |

NOTE. AIM – acceptability of Intervention Measure (AIM); IAM – Intervention Appropriateness Measure (IAM); FIM – Feasibility of Intervention Measure (FIM); FGD – focus group discussion.

## Data collection and analysis

Quantitative data were collected using the Swedish version of the Acceptability of Intervention Measure (AIM), Intervention Appropriateness Measure (IAM), and Feasibility of Intervention Measure (FIM), validated implementation outcome measures developed for use in implementation research. Each scale consists of four items rated on a five-point Likert scale, yielding total scores ranging from 4 to 20. In the original psychometric evaluation, all three measures demonstrated good to excellent internal consistency (Cronbach's α = 0.85–0.91), a unidimensional factor structure, and evidence of construct validity, supporting their use as distinct but related early-stage implementation outcomes [28]. Qualitative data were collected via focus group discussions (FGD). These data were collected in parallel and analysed simultaneously, in a convergent-parallel design (Fig 1).

**Quantitative data.** All HCWs at Unit A and B (n = 29) were invited to participate in the quantitative survey (S1 Appendix) via email and verbally in December 2021; 14 consented. Survey data were collected via a secure online platform (BASS) provided by Karolinska Institutet at three time points (1, 4, and 12 months) after intervention initiation (Fig 1). A Shapiro–Wilk test of normality was performed and indicated that data for all three outcome measures deviated significantly from a normal distribution (W = 0.734–0.860, p = .001 − .049). Therefore, descriptive statistics for each measure were presented, including median, interquartile range (IQR), and response rates. Since no established cut-offs exist for

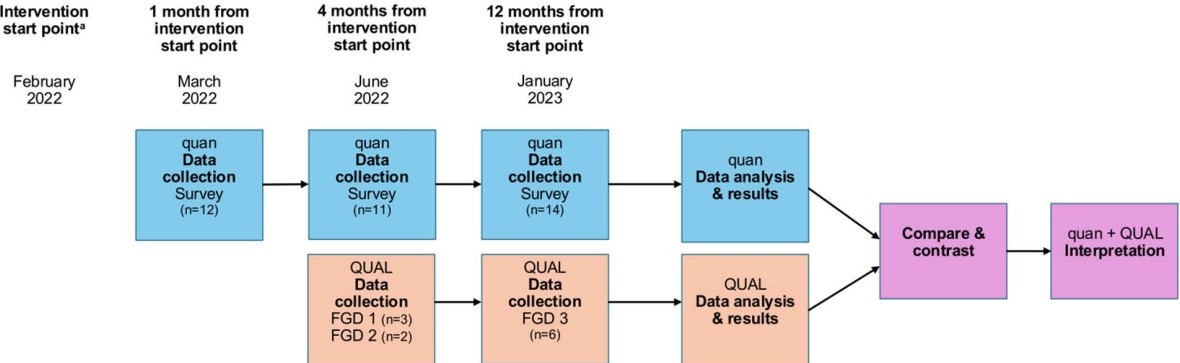

**Fig 1. Overview of mixed methods convergent-parallel design, including the number of participants at each time point. NOTE.** FGD – focus group discussion. ªBraining was launched for patients in February 2022. Patients were recruited between February and March 2022, after which they participated in the 12-week Braining intervention. From June 2022, units delivered Braining independently with the possibility of implementation support twice a year.

AIM, IAM, and FIM, we followed previous research and interpreted scores of 16 or higher (≥ 80% of the maximum score) as indicative of a "good" level of acceptability, appropriateness, or feasibility [29]. No inferential statistical tests were performed. Individual variations generally showed stable patterns over time with some individual fluctuations.

**Qualitative data.** All HCWs at Units A and B (n = 29) were invited (via email and verbally) to participate in FGDs about Braining's implementation. Only Braining team members agreed to participate (n = 8). Three semi-structured FGDs were conducted at two time points post-intervention initiation. The first two were held in June 2022 with three and two participants (due to a last-minute drop-out), respectively. The third FGD was held in January 2023 with six participants, three of whom had already participated in the previous FGDs (Fig 1). FGDs were held at the units and were moderated by a registered nurse with implementation expertise but who was not otherwise involved in the project. Two semi-structured guides were developed by the research team, based on IOF and relevant literature (S2 and S3 Appendix). The guide for the third FGD incorporated insights from the earlier FGDs and focused more on adaptation and sustainability. All FGDs were audio-recorded following written and verbal consent, and transcribed verbatim for analysis.

Data were analysed using manifest and latent qualitative content analysis with an abductive approach, as described by Graneheim and Lundman [30,31]. Briefly, two coders (EAS-G, a qualitative researcher not involved in the project at the time, and RM, a doctoral student and Braining's project manager) independently read the transcripts several times to gain a sense of the whole. Meaning units were subsequently identified and coded in Dedoose using open coding guided by the IOF. Coded meaning units were then grouped into sub-categories and coalesced into categories in Excel. EAS-G and RM, in consultation with AB and SS, met regularly to discuss and reach consensus on the analysis.

**Mixed methods integration.** Quantitative and qualitative data were interpreted in parallel to understand the acceptability, appropriateness and feasibility of Braining more comprehensively, identifying commonalities and inconsistencies between findings. This approach allowed us to harness the strengths and offset the weaknesses of the two methodologies [32,33].

## Ethics approval and consent to participate

The study was approved by the Swedish Ethical Review Authority (reference numbers: 2021−04037, 2021-05728-02, 2022-00188-02, and 2022-06854-02). Ethical principles for medical research involving human subjects were adhered to in accordance with the Declaration of Helsinki. Specifically, all participants received detailed information about the study and provided written informed consent before data collection commenced. Participants were informed that their participation

was voluntary and that they could withdraw at any time without consequence. Further, managers endorsed HCWs' participation in the focus group discussions, which helped minimise stress for HCWs who had taken time off from clinical duties. Finally, all data collected was handled in accordance with the General Data Protection Regulation (GDPR).

## Results

The findings of this mixed-methods study are described below for the three key implementation outcomes of interest: acceptability, appropriateness, and feasibility of Braining (Table 2).

### Participants

Of the 29 HCWs working at the units, 16 participated in the study: 14 completed the surveys, eight participated in the FGDs, and six contributed to both. Consequently, eight participants provided survey data only, two provided qualitative data only, and six contributed to both components. Participants represented a range of professions, including physicians, rehabilitation coordinators, registered nurses, clinic managers, addiction therapists, psychologists, and psychotherapists. Participant demographics are presented in Table 3.

### Acceptability

The median of AIM ratings was generally favourable and stable across the three time points (median 16–18; 55–79% ≥ 16/20), using a prespecified 'good' threshold of ≥16/20, with some individual fluctuations (Table 4).

Acceptability of Braining was generally reported to be high among FGD participants, who expressed overall enthusiasm for offering patients a locally available PE treatment rather than simply encouraging them to exercise independently outside the healthcare system. They viewed Braining as both enjoyable and meaningful, and appreciated that Braining-related tasks provided a break from their daily routines. However, some initially doubted whether Braining could be successfully integrated into a high-pressure environment and voiced concerns that patients might be unwilling to participate. Upon reflection, they acknowledged that such initial scepticism was normal and noted that it diminished once the team started implementing Braining in practice, as also reflected in participants' AIM ratings.

The FGD participants were surprised that some colleagues outside the Braining team showed limited interest in the intervention. They had initially expected Braining to appeal more to HCWs than to patients; however, patient interest was

Table 3. Demographics of healthcare workers who participated in the survey (n = 14) and focus group discussions (n = 8).

| Healthcare worker demographics | Quantitative survey (n = 14) | Focus group discussions (n = 8) |
|---|---|---|
| **Sex**, n | | |
| Female | 12 | 5 |
| Male | 2 | 3 |
| **Age in years**, n | | |
| 25-39 | 3 | 3 |
| 40-54 | 7 | 5 |
| 55-69 | 4 | – |
| **Years of experience**, n | | |
| <5 | 2 | 1 |
| 6-9 | 3 | 1 |
| 10-19 | 3 | 4 |
| 20-29 | 4 | 2 |
| ≥30 | 2 | – |

**Table 4. Acceptability of Intervention Measure (AIM) ratings across three time points post-intervention start (n = 14).**

|  | 1 month post start (n = 12) | 4 months post start (n = 11) | 12 months post start (n = 14) |
|---|---|---|---|
| Response rate (%) | 12/14 (86) | 11/14 (79) | 14/14 (100) |
| Median (IQR) | 18.0 (15.75-20.0) | 16.0 (14-20.0) | 18.5 (16-20.0) |
| Proportion of participants rating ≥16 (%)[a] | 9/12 (75) | 6/11 (55) | 11/14 (79) |

NOTE. [a]A rating of ≥16 was deemed "good", in accordance with Watkins et al. [29].

described as generally high, with few exceptions. The participants reflected that a more engaging introduction to Braining for all HCWs might have increased interest and engagement. They also identified additional factors that may have influenced HCWs acceptability of Braining, which are described below.

**Attitudes towards PE and HCWs' own participation in PE.** The participants reported that all Braining team members valued the physical and mental health benefits of PE, enjoyed participating in PE, or had prior experience with it. The team observed that HCWs who did not regularly engage in PE themselves were generally less accepting of Braining. To increase HCWs' acceptability, the team suggested incorporating less intensive forms of PE into Braining, such as a walking group.

*"Maybe what was needed was a different form of cardio activity—like a walk—something that doesn't get so intense. It's often staff who are already a bit interested in exercise who want to push themselves and enjoy it… But if you don't have that kind of background at all, it's not that easy." (FGD 2)*

**Patient engagement and well-being.** The participants initially anticipated challenges in recruiting patients. However, they soon realised they had underestimated patient interest and were pleasantly surprised by the high level of engagement. Immediate patient feedback following training sessions enhanced their sense of purpose and motivation to implement Braining.

*"…we get immediate feedback from the sessions and from patients who feel better. That's a big advantage for us as clinicians, to feel that we're doing something good." (FGD 2)*

Braining's welcoming environment for patients with SUD, who often face societal stigma, also contributed to the participants' acceptance of the intervention. Patients informed the Braining team that they felt free to express their true selves without having to adhere to any specific expectations. Participants also observed that training alongside patients helped positively challenge existing power imbalances. According to the FGD participants, this approach made patients feel "normal" and more included in society. Nonetheless, offering PE as a treatment was reportedly not always accepted by patients. Some declined, finding the suggestion offensive, as they felt their condition was not being taken seriously and that PE was unsuitable for them.

*"It has landed differently with different patients. Some are like, "What? Is this supposed to be treatment? I want real treatment." They feel it's minimising their problems…" (FGD 1)*

**Working as a Braining team.** The participants found working within a Braining team stimulating, especially when overcoming implementation challenges. Additionally, they felt the teamwork strengthened their relationships and improved their attitudes towards Braining. However, the participants also noted that excessive enthusiasm could be a barrier to acceptability among HCWs outside the Braining team. While enthusiasm is valuable, they explained that it can sometimes

overshadow the workload and effort required for successful implementation. They believed that focusing solely on the positive aspects of Braining while ignoring concerns could negatively impact overall acceptability.

*"There's a risk that those who are very enthusiastic may sometimes downplay how demanding certain things are. There's this attitude of, "It'll work out", and while it often does, for some people, their initial impression was that the process would be less demanding than it actually was once they became involved into the details." (FGD 1)*

## Appropriateness

The median of IAM ratings was consistently favourable, with a slight dip at month 4 and subsequent increase at month 12 (median 16–19.5; 86–92% ≥ 16/20; pre-specified 'good' threshold), see Table 5.

The participants believed that Braining aligned with their mission to support patients with SUD in increasing their PE levels and was generally well-suited to their patient demographic. Some noted that they had wanted to offer PE as a treatment for some time and therefore welcomed Braining's structured approach. This positive perception was reinforced as the participants observed improvements in their patients.

*"Regarding the actual treatment method, I was 100% very positive about introducing it. I think physical activity—and helping people do things in a group—is really important for our patients." (FGD 2)*

**Braining's fit for the patient demographic and health status.** The FGD participants were surprised that patients with a heavy disease burden—who they expected would avoid PE—showed a strong interest in Braining. The program's flexibility—allowing trainers and patients to adjust exercise intensity and offering sessions both indoors and outdoors—was seen as a key factor in this engagement. This experience prompted the participants to reflect further on patients who did not meet the intervention's inclusion criteria, specifically those with cardiovascular disease, ongoing psychosis, or eating disorders. While the participants agreed with these criteria, they described a specific case in which they consulted an eating disorder clinic about a patient diagnosed with an eating disorder who was interested in participating in Braining. The clinic recommended encouraging the patient's participation but advised the team to monitor for any compensatory behaviours.

In contrast, recruiting young adults proved challenging, even though the participants initially thought this healthier group would be more likely to participate. Some younger patients already exercised independently and saw no need for additional PE, while others lacked interest. As a result, the participants questioned whether Braining was appropriate for this subgroup without adaptations to meet their specific needs.

*"[…] the challenge is getting younger patients involved. […] We really thought this would be a perfect fit for them. Or, well, maybe not a perfect fit exactly, but they're in better physical shape than the older ones. It's not so much pure*

**Table 5. Intervention Appropriateness Measure (IAM) ratings across three time points post-intervention start (n = 14).**

|  | 1 month post start (n = 12) | 4 months post start (n = 11) | 12 months post start (n = 14) |
|---|---|---|---|
| **Response rate** (%) | 12/14 (86) | 11/14 (79) | 14/14 (100) |
| **Median (IQR)** | 19.5 (16.0-20.0) | 16.0 (16.0-19.5) | 19.0 (16.0-20.0) |
| **Proportion of participants ≥16** (%)[a] | 11/12 (92) | 10/11 (91) | 12/14 (86) |

NOTE. [a]A rating of ≥16 was deemed "good", in accordance with Watkins et al. [29].

*alcohol dependence, you know—they're physically a bit more 'intact'. But no, it just hasn't taken off. Some exercise on their own already and some just feel it's awkward." (FGD 1)*

Upon reflection, the FGD participants noted that motivating younger patients to engage in any form of treatment—not only PE but also mindfulness groups and one-on-one psychological therapy—is a common challenge. They suspected that younger patients struggle to commit to long-term treatments, although they did not elaborate on the reasons. Finally, the participants believed that Braining was particularly beneficial for unemployed patients, who had more time to attend the sessions.

**Perceived changes in patients' quality of life and lifestyle behaviours.** The FGD participants described perceived improvements in patients' quality of life following participation in Braining. They shared accounts of patients reporting substantial personal benefits, which were described as profound in some cases; one patient stated that they would not have survived without Braining, while another reported that participation supported continued abstinence from substances and enabled admission to further treatment. According to the participants, patients also described Braining as supportive of broader lifestyle changes. These observations were based on the FGD participants' reflections and patient narratives shared during the intervention period.

Braining also appeared to enhance social connections among patients. For example, when Braining was paused for the summer, patients were so motivated to continue exercising that they formed an independent workout group without Braining staff. The participants found this particularly inspiring, noting how patients encouraged and supported each other through conversations before and after Braining sessions. The FGD participants specifically witnessed patients become motivational figures for their peers, and observed how these individuals grew from the experience. Additionally, some patients with social anxiety were able to expand their social networks through Braining.

*"…they encourage each other. They check in with one another while waiting for us. They walk around and chat. It seems like they are expanding their social networks. As [name] mentioned, some may have social anxiety and a very limited personal network, but when they come here, they find a sense of belonging. They support one another—some-one who has struggled a lot might become a source of positivity for someone else, and they grow through that experi-ence." (FGD 1)*

The participants also reflected on Braining's influence on patients' exercise habits. Although the team did not routinely advise patients on how to continue exercising after the 12-week programme, they observed that some patients continued to train independently post-treatment. The participants also noted that some patients who participated in Braining regularly began exploring other types of PE alongside the programme.

### Feasibility

The median scores for participants' feasibility (FIM) ratings remained relatively stable over the three time points, with minor fluctuations noted between months 1 and 12 post-intervention start (Table 6).

The participants were initially sceptical about Braining's feasibility. They were concerned that it might increase their workload without offering significant benefits to them or their patients, and they questioned whether patients would partic-ipate, potentially leading to unattended sessions. They also acknowledged the general challenge of implementing a new treatment. However, after gaining experience with Braining, the participants found the implementation effective and gener-ally feasible for routine care, though they identified some feasibility issues to address in future iterations.

**Patient recruitment and engagement in Braining.** According to the FGD participants, several factors influenced patient recruitment and engagement in Braining. Braining's visibility and flexibility, patient motivation, and availability played key roles. The method was regarded as highly visible: music from ongoing workout sessions, participants in

**Table 6. Feasibility of Intervention Measure (FIM) ratings across three time points post-intervention start (n = 14).**

|  | 1 month post start (n = 12) | 4 months post start (n = 11) | 12 months post start (n = 14) |
|---|---|---|---|
| **Response rate** (%) | 12/14 (86) | 11/14 (79) | 14/14 (100) |
| **Median (IQR)** | 17.5 (15.75-19.0) | 18.0 (15.0-20.0) | 16.0 (16.0-19.75) |
| **Proportion of participants ≥16** (%)[a] | 9/12 (75) | 7/11 (64) | 11/14 (79) |

NOTE. [a]A rating of ≥16 was deemed "good", in accordance with Watkins et al. [29].

workout attire in the corridor, and the Braining schedule displayed in the waiting room all contributed to its prominence and helped attract patients.

The participants, however, noted that patient recruitment was also influenced by HCWs' attitudes, with efforts varying among HCWs, and that most patients were recruited by the Braining team. They believed it was easier for Braining team members to present the treatment and encourage participation, as they were so immersed in Braining. The participants also emphasised that starting with a medical assessment increased their confidence in recruiting patients; screening for health risks and consulting a physician when needed, helped them feel secure in the process.

One effective recruitment strategy was inviting patients to participate in a workout session alongside HCWs. This approach was seen as lowering the threshold for participation by offering a natural, shared entry point. Above all, the team found that the best recruiters were the patients themselves, as enrolled patients often inspired others to join. The participants also noted that HCWs' persistence in encouraging participation was pivotal for some patients. For example, one patient expressed appreciation for the participants' continued efforts, considering their persistence a key factor in their decision to join Braining.

*"[…] I'd suggest, "Hey, why don't you come along next time?" and they'd say no. Then maybe after four tries, I'd get them to join by saying, "Let's go together." It was a very natural way to involve patients—just saying, "Put on your sneakers and we'll go together." That's how I got patients to join. […] I remember someone said, "Thanks for pushing me—now I feel so much better." So it wasn't just that I mentioned it once." (FGD 3)*

The FGD participants noted that some patients needed repeated encouragement after enrolment. Motivating patients before Braining workouts via phone or text messages served as reminders and helped increase patients' sense of security. However, participants found individual check-ins challenging. While the team generally observed higher participation after reaching out to patients the day before workout sessions, some patients avoided pre-booking to avoid these calls. These patients felt the individual check-ins were unnecessary, believing they could decide for themselves whether they were fit to participate. Additionally, the participants described the patient group as spontaneous and often unstructured, with many not answering phone calls even after booking. Finally, patients' daytime commitments and the units' limited opening hours were also reported to hinder participation in Braining.

**Managing Braining's complexity and structure through staff collaboration.** The participants described the intervention as complex, involving weekly team meetings, training schedules, and patient visits—all of which require significant time and commitment. They emphasised that structured weekly meetings with a strict, predefined agenda were essential for Braining's feasibility. These meetings allowed the team to track progress, coordinate workout schedules, oversee patient care, and monitor both patient and staff participation. The meeting also provided an opportunity to identify patient non-compliance and implement corrective measures to ensure adherence.

*"We have weekly meetings with a very clear and strict agenda… Like, 'Have we held the sessions? Were they at least 30 minutes long? Have we followed the method?' […] The fact that they are part of the method is great. It's a great way*

*to stay organised. If we had less structured meetings, I think there would be a risk that things suddenly don't work, like forgetting who is supposed to lead a session. We've drifted a bit sometimes and started to slack in certain areas, but we've flagged that during meetings." (FGD 1)*

The participants also reported that Braining improved their work environment by fostering collaboration across the two units. Although staff belonged to separate but adjacent units, they had previously interacted very little. Braining changed this, promoting both professional and social engagement. Unit B even noted that this increased collaboration influenced management's decision to continue working with Braining. Strengthened relationships through informal conversations and frequent joint meetings also led to an improved referral system.

*"That's one of the reasons we were able to keep working with Braining […] because we have improved collaboration between clinics as a result […] so it's a very positive organisational outcome. […] It's not just that we know each other and chat over coffee—we've started working more closely on referrals, patient care, unit meetings, and many other concrete collaborative efforts. I think Braining has contributed to this […] since we handle referrals together, it has completely changed how we collaborate […]. There are never any issues. No 'us vs. them'—which is great." (FGD 3)*

**Integration into existing workflows and routine care.** The participants described having successfully integrated Braining into their existing workflows. A central part of their routine was introducing Braining during the patient's initial consultation with a medical doctor, who explained how the intervention could be incorporated into the patient's rehabilitation plan. Many individual check-ins before Braining sessions were combined with routine procedures, such as blood and urine sampling. Additionally, the participants suggested that Braining could complement other treatments, such as psychological therapy for anxiety disorders, by providing a safe environment with access to HCWs during Braining sessions.

*"[…] It becomes part of the exposure—to go and exercise in a safe environment, like with a nurse or a psychologist. Because it's us who lead these sessions." (FGD 1)*

One year after implementation initiation, Braining was regarded as a natural part of the treatment offerings at the units. FGD participants described Braining as a meaningful and energising part of their daily work, stating that their range of treatment would feel incomplete without it. On challenging days, staff reported arriving at work feeling drained yet leaving invigorated after participating in a Braining session.

*"It's an important part, very energising, I think. I agree, it feels like a natural component when we present our treatment options—having a variety of approaches. Now, it would feel lacking to offer only discussion groups. Some days, as you said, when it comes to being energising, you might walk in feeling a bit heavy. But after being there—you all know how it is—you almost float out of the room. It's an incredible experience." (FGD 3)*

**Managerial support and local champions.** The FGD participants described high engagement and strong support from management. Unlike other treatments, the manager viewed Braining not only as a patient intervention but also as a staff benefit, highlighting its ability to add variety to the workday and provide the positive effects of PE. The participants also reflected on whether Braining's successful implementation was due to the intervention's structure or to the dedication of highly invested team members. Initially, there were concerns about relying too heavily on a few local champions, fearing the programme might falter if someone resigned. However, after one year, the team noted that membership had actually increased, creating a more stable foundation. They expressed confidence in Braining's sustainability, stating that its continuity would not be threatened by individual resignations.

*"I think there are more dedicated and enthusiastic people now. […] Yes, it was more fragile in the beginning when everything was just getting started. […] Now the routine is established, and we are 10 people. …I was thinking of a former colleague who was involved. Even though she was very engaged, it still worked out after she left." (FGD 3)*

Finally, participants suggested organising annual meetings or a conference for all Braining teams across units offering the service, to help maintain Braining and prevent it from dissolving.

## Discussion

Guided by Proctor's Implementation Outcomes Framework (IOF), our findings show that participants perceived Braining as acceptable, appropriate, and feasible for patients with SUD. Quantitative ratings (AIM, IAM, FIM) remained above the interpretive threshold (≥16/20) for most participants throughout the 12-month period. These findings converged with the qualitative data, which indicated that Braining was perceived as acceptable over time, characterised by strong patient enthusiasm, a clear yet adaptable structure, and support from a stable team and engaged leadership. However, variable staff engagement, resource-intensive motivational efforts, and challenges in reaching younger adults emerged as key barriers. In contrast, organisational flexibility, structured routines, joint exercise sessions, and integration into routine care functioned as important facilitators.

While IOF predict potential implementation success [25], they do not help explain anticipated or actual implementation outcomes. Determinant frameworks, such as the Consolidated Framework for Implementation Research (CFIR), can help delineate contextual factors that influence the outcomes of implementation efforts [34,35]. We therefore structure our discussion around CFIR to elucidate our findings, focusing on setting-level barriers and facilitators that may impact implementation success and to inform future adaptations.

### Innovation domain

Within CFIR, the Innovation domain encompasses core attributes of an intervention, such as design, complexity, adaptability, and perceived relative advantage. We apply this domain to describe the intervention Braining and therefore use the term innovation to examine how its characteristics may shape perceptions of acceptability, appropriateness, and feasibility.

Initial scepticism among some participants diminished as Braining embedded in practice, with acceptability and feasibility dipping temporarily at 4 months before rising by 12 months. This suggests that HCWs needed time and experience to recognise Braining's relative advantage over traditional brief physical activity counselling through direct patient experience.

The FGD participants reported high patient enthusiasm, which may have increased acceptability and influenced colleagues' attitudes beyond the Braining team. Joint staff-patient exercise sessions may therefore represent an important motivational mechanism, as they allow HCWs to directly observe reported patient improvements and assess the innovation in real-world settings. This is consistent with findings from group-based interventions in psychiatric care [36] and may have strengthened Braining's perceived value and fit.

The innovations' clear structure, predefined procedural elements, and embedded motivational elements were repeatedly highlighted as features that reduced perceived complexity and made delivery easier than initially anticipated. At the same time, its flexible format, including the possibility to adapt exercise intensity and modality to individual patients, appeared to support both trialability and adaptability, enabling the Braining team to "test" and refine delivery in ways that fit their settings and patients.

### Inner setting domain

Systematic reviews and meta-analyses confirm the potential of PE as an effective adjunct to SUD treatment, improving physical and mental health outcomes [37–39]. FGDs echoed this, with participants describing Braining as mission-aligned, despite initial concerns about suitability for individuals with a high disease burden. This mission alignment supported

perceptions of appropriateness and was reflected in consistently favourable quantitative ratings across all timepoints. High participation among these patients indicates that Braining's flexibility and adjustable intensity enable broad inclusion, consistent with evidence that individualised innovations enhance adherence in SUD populations [12,38,40].

At the 12-month follow-up, participants described Braining as integrated into routine practice at participating clinics, indicating its relative priority, mission alignment, and compatibility were embedded in everyday work. This reflects local routinisation rather than formal adoption at the organisational or system level, neither of which was evaluated. Participants identified factors for continued delivery: a stable multidisciplinary Braining team, strong managerial endorsement, and the incorporation of tasks into existing workflows. These align with recommendations for sustaining lifestyle interventions in mental health care through organisational prioritisation, leadership support, and service integration [39]. However, adoption and longer-term sustainability change remain to be examined.

### Individuals domain

Patients' opportunity to participate in Braining was reported by participants to depend on HCWs' motivation to offer the innovation, underscoring how individual clinicians' attitudes shape implementation success, as reflected in prior research on exercise interventions in mental health services [41,42]. Acceptability was enhanced by HCWs' own positive experiences with PE; those familiar or comfortable with PE were more likely to promote Braining, whereas HCWs with less experience or confidence in PE were less engaged. This mirrors findings identifying role ambiguity, lack of confidence and safety concerns as barriers to staff-delivered PE interventions [41,43].

Further, HCWs' assumptions about patients' motivation or capability to engage in Braining may unintentionally risk inequitable access, particularly for individuals with complex psychiatric or physical comorbidities. Clinician scepticism regarding patients' interest, understanding, or safety in engaging in lifestyle interventions has previously been identified as a barrier to equitable access for this group [39]. Braining's pre-enrolment medical assessments reassured FGD participants and supported safer decision-making, aligning with recommendations for PE in medically complex SUD populations [38,39]. Increasing use of patient-centred shared decision-making [44] may support equitable access by ensuring that patient preferences, rather than HCWs' assumptions, guide participation.

Survey data confirmed Braining's high appropriateness, yet recruiting younger adults proved challenging. Many already exercised independently or preferred autonomy over structured programmes, also suggesting tailored approaches that emphasise variety and flexibility to better align with their needs [38,45].

Participants described the structured motivational components of the 12-week programme as a central facilitator to both recruitment and sustained participation. Frequent encouragement, regular brief check-ins, and personalised invitations were perceived as effective strategies for initiating patient participation, consistent with earlier research on exercise motivation among patients [8,41,46]. Brief affirmations and positive reinforcement during Braining sessions further supported ongoing participation and enhanced the intervention's appeal.

Nevertheless, motivational work was also described as resource-intensive and raising concerns about sustainability and equitable delivery. Some patients, particularly those hesitant or facing motivational barriers, were described as needing repeated outreach, including phone calls or direct invitations. Although regular check-ins were beneficial for many patients, others found them unnecessary, again underscoring the importance of assessing needs to balance motivational support with patient autonomy. These tensions between benefits, burden, and sustainability have also been described in studies of lifestyle and behaviour-change interventions, where intensive support improves uptake but may be difficult to maintain at scale [39].

### Implementation process

A core Braining component is the multidisciplinary implementation team, which convenes regular meetings with a pre-defined agenda to ensure coordinated delivery. Participants emphasised this team's pivotal role in both initiating and sustaining the innovation through teaming, in which shared responsibility and stable processes enable consistent

implementation. Early concerns about over-reliance on individual champions in the team highlighted vulnerabilities when success depends on isolated individuals. However, over time, the team stabilised with committed members contributing to delivery, reducing champion dependency and strengthening Braining's unit position [47].

The structure of regular meetings appeared to support the team in assessing needs (patient-level and organisational challenges) and context (local workflows, resources, unit priorities), planning (coordinated recruitment, delivery, and barrier management), and tailoring strategies (procedure adjustments based on insights). These forums also enabled reflection and evaluation through progress reviews and discussions of barriers, which may have supported ongoing adaptation processes. Such processes are considered important for sustaining implementation in dynamic clinical settings [35].

## Innovation outcomes

The FGD participants perceived benefits in patients' quality of life and lifestyle following participation in Braining. They reported that some patients credited the innovation with profound personal and social benefits, with the social value particularly beneficial for individuals with social anxiety, as it enabled them to expand their networks and develop a sense of belonging. These observations are consistent with systematic reviews showing that exercise innovations can improve quality of life, mood, and social functioning in people with SUD, particularly in group-based formats that foster social support [8,12,38–40]. Importantly, although PE has been associated with improvements in craving and withdrawal symptoms [37,48], effects on alcohol and other SUD outcomes are heterogeneous across studies, with some trials and meta-analyses reporting limited or no impact on consumption or binge drinking [49,50]. It may be contingent on how the innovation is designed and implemented. Consequently, while Braining appears to offer meaningful personal and social benefits, conclusions regarding its causal impact on patients' substance use and broader lifestyle outcomes cannot yet be drawn; further investigation is needed.

Moreover, Braining appeared to positively influence the participants' perceived sense of purpose and enjoyment at work, raising the question of whether the innovation may also affect job satisfaction, work engagement, and perceived value among HCWs. These observations reflect participant-reported perceptions rather than formally measured outcomes. Such outcomes are particularly relevant given evidence that higher staff engagement is associated with reduced turnover and improved workforce productivity in healthcare settings [51].

An additional and unanticipated outcome, as reported by participants, was enhanced collaboration between the two units. According to participants, Braining strengthened peer-to-peer relationships and improved referral processes, suggesting that innovation may confer organisational benefits beyond patient-level effects. Such improvements in collaboration and coordination are consistent with implementation literature emphasising the role of organisational context and interprofessional collaboration in successful implementation [35]. Taken together, these participant-reported findings highlight that HCWs may experience distinct, often overlooked outcomes, such as satisfaction and work experience with delivering an innovation. As emphasised within the Quadruple Aim framework, improving the work-life and well-being of HCWs is a critical consideration, as their experiences may influence the sustainability of innovations in real-world practice [52].

## Future directions

Braining aligns well with outpatient psychiatric care for SUD. Still, refinements could enhance equitable reach and sustainability, such as strategies targeting younger adults, low-intensity exercise options (e.g., walking groups, yoga) to boost HCW and patient engagement, brief skills training to standardise delivery, and a learning collaborative to maintain Braining teams post-scale-up. Combining Braining with psychological interventions may further improve outcomes for patients with depression or social anxiety [42,53].

Future studies should test the effectiveness and implementation requirements of these refinements, including integrated care models, while assessing HCWs' satisfaction, implementation fidelity, long-term sustainability, costs, and cost-effectiveness during service scale-up.

 

## Strengths and limitations

A key strength is the mixed-methods design, which provides nuanced insights into Braining's real-world implementation in Swedish outpatient psychiatry over 12 months and serves as a formative evaluation prior to an RCT. Although all staff were invited to participate, limitations include a small sample size and low HCW participation (qualitative data were solely from the Braining team), which restricts the transferability of qualitative findings and the generalisability of quantitative results. While the findings elucidate implementation factors, they cannot address causal or mechanistic effects; a process evaluation or fidelity assessment would be required. Results thus reflect perceived relevance rather than demonstrated effectiveness of Braining.

## Conclusions

Braining was perceived by participants as acceptable, appropriate, and feasible for patients with SUD in outpatient psychiatry. Perceived routinisation was associated with a structured yet adaptable innovation design, team-based implementation processes, and supportive organisational conditions, including leadership endorsement and cross-unit collaboration. As a formative mixed-methods evaluation, these findings provide insight into implementation processes and perceived relevance rather than demonstrated effectiveness.

At the same time, variability in HCWs' engagement and challenges in reaching younger patients indicate the need for targeted implementation strategies to support equitable reach and long-term sustainability. Overall, the findings suggest that Braining shows promise for supporting patient engagement in PE in outpatient psychiatry and warrant further investigation as it is scaled across services.

## Supporting information

**S1 Appendix. Quantitative survey AIM, IAM and FIM.**
(DOCX)

**S2 Appendix. Focus group discussion guide 1.**
(DOCX)

**S3 Appendix. Focus group discussion guide 2.**
(DOCX)

**S4 Appendix. List of abbreviations.**
(DOCX)

## Acknowledgments

We would like to express our sincere gratitude to all the staff who took the time to participate in this study, to the Braining team and managers at the two units, and, in particular, to the lead coordinators who supported Braining's local implementation.

## Author contributions

**Conceptualization:** Åsa Anger, Carl Johan Sundberg, Lina Martinsson, Anna Bergström, Sigrid Salomonsson.

**Data curation:** Rebecka Mac, Anna Bergström, Sigrid Salomonsson.

**Formal analysis:** Rebecka Mac, Erika A Saliba-Gustafsson, Anna Bergström, Sigrid Salomonsson.

**Investigation:** Rebecka Mac.

**Methodology:** Rebecka Mac, Anna Bergström, Sigrid Salomonsson.

**Project administration:** Rebecka Mac.

**Supervision:** Rebecka Mac, Anna Bergström, Sigrid Salomonsson.

**Visualization:** Rebecka Mac, Erika A Saliba-Gustafsson, Sigrid Salomonsson.

**Writing – original draft:** Rebecka Mac, Erika A Saliba-Gustafsson, Anna Bergström, Sigrid Salomonsson.

**Writing – review & editing:** Rebecka Mac, Erika A Saliba-Gustafsson, Åsa Anger, Carl Johan Sundberg, Lina Martinsson, Anna Bergström, Sigrid Salomonsson.

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
