## [Decision Letter · Decision Letter 0]

28 Dec 2025

PONE-D-25-52478Moving Psychiatry Forward – Implementation and evaluation of Braining, a structured physical exercise intervention in outpatient psychiatry: a convergent-parallel mixed methods studyPLOS One

Dear Dr. Mac,

Thank you for submitting your manuscript to PLOS ONE. After careful consideration, we feel that it has merit but does not fully meet PLOS ONE’s publication criteria as it currently stands. Therefore, we invite you to submit a revised version of the manuscript that addresses the points raised during the review process.

We look forward to receiving your revised manuscript.

Kind regards,

Anthony A. Olashore, MBCHB, PhD.

Academic Editor

PLOS One

Journal Requirements:

Reviewers' comments:

Reviewer's Responses to Questions

**Comments to the Author**

1. Is the manuscript technically sound, and do the data support the conclusions?

Reviewer #1: Yes

Reviewer #2: Yes

2. Has the statistical analysis been performed appropriately and rigorously? 

Reviewer #1: Yes

Reviewer #2: Yes

3. Have the authors made all data underlying the findings in their manuscript fully available?

Reviewer #1: Yes

Reviewer #2: No

4. Is the manuscript presented in an intelligible fashion and written in standard English?

Reviewer #1: Yes

Reviewer #2: Yes

5. Review Comments to the Author

Reviewer #1: Thank you for the opportunity to review this manuscript. This is a well-written and conceptually strong mixed-methods implementation study evaluating the acceptability, appropriateness, and feasibility of the “Braining” physical exercise intervention within Swedish outpatient psychiatry. The manuscript is clear, methodologically transparent, and presents a coherent integration of qualitative and quantitative findings. It makes an important contribution to implementation research in mental health—an area where evidence-based physical exercise interventions remain underutilized.

That said, several areas could be strengthened to enhance the manuscript

1. Statistical analysis of normality

The authors note that both mean and median values were calculated due to the small sample size and potential skewness. This issue could be addressed more rigorously by performing a formal test of normality (e.g., Kolmogorov–Smirnov or Shapiro–Wilk test). Reporting this would provide stronger justification for the use of descriptive statistics and interpretation of central tendency.

2. Discussion

a. The discussion would benefit from substantial revision to reduce redundancy and deepen analytical interpretation. Several themes—such as initial staff skepticism followed by enthusiasm, positive patient responsiveness, and difficulty engaging younger patients—are repeated multiple times across sections. Condensing these will improve readability and allow more space to discuss the implications of findings within implementation science frameworks.

b. The authors identify motivational work as a key component of Braining’s success. While plausible, the current study design does not allow for conclusions about causal or mechanistic effects. A process evaluation or implementation fidelity assessment would be required to identify which components of Braining are essential to its success. The discussion should be revised to reflect this limitation.

c. The statement “HCWs observed notable improvements in patients’ quality of life and lifestyle behaviours” should be clarified. It is not clear how these improvements were measured or defined. “Quality of life” typically requires a validated, operationalized measure (e.g., WHOQOL-BREF, SF-36). If this was anecdotal or based on qualitative impressions, it should be presented as such.

d. The claim that “structured weekly meetings played a crucial role in sustaining Braining’s implementation” is interesting but cannot be substantiated by this study design. The authors may report that participants perceived these meetings as important, but attributing causal impact requires a process or outcome evaluation. Please revise this phrasing accordingly.

e. The study was designed to evaluate acceptability, appropriateness, and feasibility of Braining, not to determine which components of Braining drive effectiveness. The discussion should explicitly acknowledge this scope and avoid making inferences about which aspects (e.g., teamwork, motivation) are “key” to success without supporting evidence.

f. The discussion would benefit from clarification on whether Braining has now been adopted as part of standard care in the participating clinics. What mechanisms have been put in place to ensure this?

Reviewer #2: Moving Psychiatry Forward – Implementation and evaluation of Braining, a structured

physical exercise intervention in outpatient psychiatry: a convergent-parallel mixed

methods study

This research examined an important issue that will add valuable insights to the body of knowledge. The methodology is sound, and the paper is well written with its strengths and limitations well captured. I have few observations which are highlighted:

Title: While the paper provides excellent insight into implementation of structured physical exercise in outpatient psychiatry clinics, adding the caption of “moving psychiatry forward” is framed too broadly than the scope of the study. Consider revising the title to accurately reflect the study’s boundaries.

Methodology: It will be helpful to include the psychometric properties of the validated instruments used {AIM – acceptability of Intervention Measure (AIM); IAM – Intervention Appropriateness Measure (IAM); FIM – Feasibility of Intervention Measure (FIM)}.

Results

This statement in line 228 to 231 appears complex and difficult to understand. The statement suggests that those outside the Braining Team assume the intervention will target staff rather than patients and on the contrary patient’s interest in Braining was perceived as high….

Discussion: Line 584 to 681- This statement here seems to be outside the scope of what this study addressed, with no clue from the methodology or result pointing to this.

6. PLOS authors have the option to publish the peer review history of their article (what does this mean?). If published, this will include your full peer review and any attached files.

Reviewer #1: **Yes:**Temitope Ogundare, MD, MPH

Reviewer #2: **Yes:**Julianah Mosanya

---

## [Author Response · Author response to Decision Letter 1]

19 Feb 2026

Response letter

We would like to thank the reviewers for their thoughtful and constructive comments, and the editor for the opportunity to revise our manuscript. We have carefully considered each comment and made every effort to address them thoroughly and appropriately. Below, we provide a point-by-point response to all reviewer comments. A revised version of the manuscript with changes marked in yellow is enclosed for ease of review.

Reviewer #1:

Thank you for the opportunity to review this manuscript. This is a well-written and conceptually strong mixed-methods implementation study evaluating the acceptability, appropriateness, and feasibility of the “Braining” physical exercise intervention within Swedish outpatient psychiatry. The manuscript is clear, methodologically transparent, and presents a coherent integration of qualitative and quantitative findings. It makes an important contribution to implementation research in mental health—an area where evidence-based physical exercise interventions remain underutilized.

That said, several areas could be strengthened to enhance the manuscript

Response to reviewer 1: We thank the reviewer for their positive and encouraging assessment of our manuscript. We have carefully considered the suggestions for improvement outlined below and have revised the manuscript accordingly.

1. Statistical analysis of normality

The authors note that both mean and median values were calculated due to the small sample size and potential skewness. This issue could be addressed more rigorously by performing a formal test of normality (e.g., Kolmogorov–Smirnov or Shapiro–Wilk test). Reporting this would provide stronger justification for the use of descriptive statistics and interpretation of central tendency.

Response: Thank you for your valuable comment regarding the statistical approach and data distribution. We have now performed a Shapiro–Wilk test of normality, which indicated that data for all outcome measures deviated significantly from a normal distribution (W = 0.734-0.860, p =.001 - .049). Consequently, we chose not to report means and standard deviations in the descriptive statistics. Instead, medians and interquartile ranges (IQRs) are presented. This information has been added to ease interpretation of the descriptive statistics.

See revisions in manuscript:

Line 151: A Shapiro–Wilk test of normality was performed and indicated that data for all three outcome measures deviated significantly from a normal distribution (W = 0.734-0.860, p =.001 - .049). Therefore, descriptive statistics for each measure were presented, including median, interquartile range (IQR), and response rates.

Line 209: Acceptability

Line 273: Appropriateness

Line 343: Feasibility

2. Discussion

a. The discussion would benefit from substantial revision to reduce redundancy and deepen analytical interpretation. Several themes—such as initial staff skepticism followed by enthusiasm, positive patient responsiveness, and difficulty engaging younger patients—are repeated multiple times across sections. Condensing these will improve readability and allow more space to discuss the implications of findings within implementation science frameworks.

Response: We thank the reviewer for this constructive and helpful comment. We have substantially revised the Discussion to reduce redundancy and strengthen analytical interpretation. Repeated themes, such as initial staff scepticism followed by increased engagement, positive patient responsiveness, and challenges in engaging younger patients, have been consolidated across sections and domains to improve readability and coherence. This restructuring has allowed us to focus more clearly on the implications of the findings and to deepen the analysis by explicitly situating the results within an implementation science framework. In particular, we have applied CFIR constructs to interpret the observed implementation outcomes.

In line with this strengthened analytical focus, we have also revised the study aim to clarify that the study sought not only to explore but also to explain healthcare workers’ perceptions of acceptability, appropriateness, and feasibility. This change reflects the more explicit use of implementation theory to interpret and contextualise the findings.

See full discussion from line 463.

Se revised aim on line 30 and 99:

This study sought to explore and explain healthcare workers’ acceptability, appropriateness, and feasibility of Braining in an outpatient psychiatric setting to inform future implementation efforts.

Response to comments b-e: We thank the reviewer for these related comments. We have carefully reviewed the manuscript and revised the wording throughout to clarify that the reported findings reflect participants’ observed experiences and perceptions, rather than inferred causal relationships. Statements that could previously be interpreted as implying causality have been rephrased to explicitly describe experiential or observational accounts. These clarifications have been applied consistently across all examples raised in comments b–e and are indicated in the revised manuscript.

b. The authors identify motivational work as a key component of Braining’s success. While plausible, the current study design does not allow for conclusions about causal or mechanistic effects. A process evaluation or implementation fidelity assessment would be required to identify which components of Braining are essential to its success. The discussion should be revised to reflect this limitation.

Response: We thank the reviewer for this important clarification. We agree that the current study design does not allow conclusions regarding causal or mechanistic effects of specific Braining components. We have therefore revised the Discussion to explicitly frame motivational work as perceived by staff rather than as a demonstrated driver of outcomes. In addition, we have added a clear methodological limitation, noting that a process evaluation or implementation fidelity assessment would be required to identify which components of Braining are essential for patient engagement and retention. These revisions are reflected in the updated Discussion section Individuals domain:

Line 505:

At the 12-month follow-up, HCWs described Braining as integrated into routine practice at participating clinics, indicating its relative priority, mission alignment, and compatibility were embedded in everyday work. This reflects local routinisation rather than formal adoption at the organisational or system level, which was not evaluated. Participants identified factors for continued delivery: a stable multidisciplinary Braining team, strong managerial endorsement, and the incorporation of tasks into existing workflows. These align with recommendations for sustaining lifestyle interventions in mental health care through organisational prioritisation, leadership support, and service integration (39). However, adoption and longer-term sustainability change remain to be examined.

c. The statement “HCWs observed notable improvements in patients’ quality of life and lifestyle behaviours” should be clarified. It is not clear how these improvements were measured or defined. “Quality of life” typically requires a validated, operationalized measure (e.g., WHOQOL-BREF, SF-36). If this was anecdotal or based on qualitative impressions, it should be presented as such.

Response: We thank the reviewer for this important comment. We agree that the original wording could be interpreted as implying intervention effects beyond the scope of the study. We have therefore revised both the Results and Discussion sections to clarify that these findings reflect HCWs’ observations and interpretations of patient-reported experiences, rather than demonstrated effects of Braining. In the Discussion, we have further framed these observations within the construct of appropriateness, emphasizing perceived relevance and fit for the patient group rather than causal outcomes. All related statements have been revised accordingly.

Line 41 Results in Abstract: Qualitative findings confirmed HCWs' positive perceptions of patient suitability, workflow integration, interprofessional collaboration, and perceived improvements in patients' quality of life and exercise motivation.

Line 316 Results:

Perceived changes in patients’ quality of life and lifestyle behaviours

HCWs described perceived improvements in patients’ quality of life following participation in Braining. Staff shared accounts of patients reporting substantial personal benefits, which were described as profound in some cases; one patient stated that they would not have survived without Braining, while another reported that participation supported continued abstinence from substances and enabled admission to further treatment. According to HCWs, patients also described Braining as supportive of broader lifestyle changes. These observations were based on staff reflections and patient narratives shared during the intervention period.

Line 566 Discussion:

HCWs perceived benefits in patients’ quality of life and lifestyle following participation in Braining. They reported that some patients credited the programme with profound personal and social benefits, with the social value particularly beneficial for individuals with social anxiety, as it enabled them to expand their networks and develop a sense of belonging. These observations are consistent with systematic reviews showing that exercise interventions can improve quality of life, mood, and social functioning in people with SUD, particularly in group-based formats that foster social support (8,12,38–40). Importantly, although PE has been associated with reductions in consumption, cravings and withdrawal symptoms (37,49,50), evidence regarding effects on substance use outcomes remains mixed. It may be contingent on how the intervention is designed and implemented. Consequently, while Braining appears to offer meaningful personal and social benefits, conclusions regarding its causal impact on patients’ substance use and broader lifestyle outcomes cannot yet be drawn; further investigation is needed.

d. The claim that “structured weekly meetings played a crucial role in sustaining Braining’s implementation” is interesting but cannot be substantiated by this study design. The authors may report that participants perceived these meetings as important, but attributing causal impact requires a process or outcome evaluation. Please revise this phrasing accordingly.

Response: We thank the reviewer for this important clarification. We agree that the study design does not allow conclusions regarding causal effects of structured weekly meetings. We have therefore revised the wording to explicitly reflect participants’ perceptions of these meetings as important for implementation, rather than attributing a causal role. The revised phrasing is now framed in experiential terms and no longer implies mechanistic impact:

Line 551: A core Braining component is the multidisciplinary implementation team, which convenes regular meetings with a predefined agenda to ensure coordinated delivery. Participants emphasised this team's pivotal role in both initiating and sustaining the innovation through teaming, where shared responsibility and stable processes enable consistent implementation.

e. The study was designed to evaluate acceptability, appropriateness, and feasibility of Braining, not to determine which components of Braining drive effectiveness. The discussion should explicitly acknowledge this scope and avoid making inferences about which aspects (e.g., teamwork, motivation) are “key” to success without supporting evidence.

Response: We thank the reviewer for this important clarification. We have revised the Discussion to more explicitly acknowledge the study’s scope, emphasising that the evaluation focused on acceptability, appropriateness, and feasibility rather than on identifying causal or mechanistic drivers of effectiveness. Statements that could be interpreted as implying that specific components of Braining were “key” to success have been rephrased to reflect perceived relevance and associations rather than demonstrated effects. We have also clarified that determining which components contribute most strongly to outcomes would require dedicated process evaluations or effectiveness studies.

See full discussion and Line 610:

While the findings elucidate implementation factors, they cannot address causal or mechanistic effects; a process evaluation or fidelity assessment would be required. Results thus reflect perceived relevance rather than demonstrated effectiveness of Braining.

f. The discussion would benefit from clarification on whether Braining has now been adopted as part of standard care in the participating clinics. What mechanisms have been put in place to ensure this?

Response: We thank the reviewer for this important clarification. We have revised the Discussion to explicitly state that healthcare workers described Braining as integrated into routine practice at the participating clinics at the time of follow-up, while clarifying that formal adoption as standard care and long-term sustainability were not evaluated in this study. We also describe the locally perceived mechanisms supporting continued delivery, including a stable multidisciplinary team, managerial endorsement, and integration of Braining-related tasks into existing clinical workflows.

Line 505:

At the 12-month follow-up, HCWs described Braining as integrated into routine practice at participating clinics, indicating its relative priority, mission alignment, and compatibility were embedded in everyday work. This reflects local routinisation rather than formal adoption at the organisational or system level, which was not evaluated. Participants identified factors for continued delivery: a stable multidisciplinary Braining team, strong managerial endorsement, and the incorporation of tasks into existing workflows. These align with recommendations for sustaining lifestyle interventions in mental health care through organisational prioritisation, leadership support, and service integration (39). However, adoption and longer-term sustainability change remain to be examined.

Reviewer #2:

Moving Psychiatry Forward – Implementation and evaluation of Braining, a structured

physical exercise intervention in outpatient psychiatry: a convergent-parallel mixed

methods study

This research examined an important issue that will add valuable insights to the body of knowledge. The methodology is sound, and the paper is well written with its strengths and limitations well captured. I have few observations which are highlighted:

Response to reviewer 2: We thank the reviewer for their positive and thoughtful evaluation of our manuscript. We appreciate the recognition of the study’s relevance, methodological rigor, and balanced presentation of strengths and limitations. Below, we address the reviewer’s observations point by point and describe how the manuscript has been revised accordingly.

1. Title: While the paper provides excellent insight into implementation of structured physical exercise in outpatient psychiatry clinics, adding the caption of “moving psychiatry forward” is framed too broadly than the scope of the study. Consider revising the title to accurately reflect the study’s boundaries.

Response: We thank the reviewer for this helpful comment. We agree that the original title may have framed the study's scope too broadly. To better reflect the study’s boundaries while retaining a descriptive and engaging phrasing, we have revised the title to:

Line 1: Getting psychiatry on the move – Implementation and evaluation of Braining, a structured physical exercise intervention in outpatient psychiatry: a convergent-parallel mixed methods study.

2. Methodology: It will be helpful to include the psychometric properties of the validated instruments used {AIM – acceptability of Intervention Measure (AIM); IAM – Intervention Appropriateness Measure (IAM); FIM – Feasibility of Intervention Measure (FIM)}.

Response: We thank the reviewer for this suggestion. We have now added descriptions of the psychometric properties of AIM, IAM, and

---

## [Decision Letter · Decision Letter 1]

19 Mar 2026

PONE-D-25-52478R1Getting psychiatry on the move – Implementation and evaluation of Braining, a structured physical exercise intervention in outpatient psychiatry: a convergent-parallel mixed methods studyPLOS One

Dear Dr. Mac,

Thank you for submitting your manuscript to PLOS ONE. After careful consideration, we feel that it has merit but does not fully meet PLOS ONE’s publication criteria as it currently stands. Therefore, we invite you to submit a revised version of the manuscript that addresses the points raised during the review process.

We look forward to receiving your revised manuscript.

Kind regards,

Anthony A. Olashore, PhD.

Academic Editor

PLOS One

Journal Requirements:

Reviewers' comments:

Reviewer's Responses to Questions

**Comments to the Author**

1. If the authors have adequately addressed your comments raised in a previous round of review and you feel that this manuscript is now acceptable for publication, you may indicate that here to bypass the “Comments to the Author” section, enter your conflict of interest statement in the “Confidential to Editor” section, and submit your "Accept" recommendation.

Reviewer #1: (No Response)

Reviewer #2: All comments have been addressed

2. Is the manuscript technically sound, and do the data support the conclusions?

Reviewer #1: Partly

Reviewer #2: Yes

3. Has the statistical analysis been performed appropriately and rigorously? 

Reviewer #1: Yes

Reviewer #2: Yes

4. Have the authors made all data underlying the findings in their manuscript fully available?

Reviewer #1: Yes

Reviewer #2: No

5. Is the manuscript presented in an intelligible fashion and written in standard English?

Reviewer #1: Yes

Reviewer #2: Yes

6. Review Comments to the Author

Reviewer #1: I thank the authors for their efforts and careful revision of the manuscript. The current draft is substantially improved compared with the previous version. However, several issues still need to be addressed.

Line 162: “Only Braining team members agreed to participate (n=8).” Please ensure that the reporting of the qualitative results and the Discussion consistently reflect this. Using the term “HCWs” throughout may be misleading regarding who contributed qualitative data. You note this in the Limitations, which is helpful; however, the wording in Results/Discussion should also be consistently aligned.

Lines 201–202: “Of the 29 HCWs working at the units, a total of 16 participated in this study. Specifically, 14 HCWs completed the surveys, of whom 8 also participated in the FGDs.” This description is unclear as written. Please clarify the total number of participants.

Lines 551–553: The claim that check-ins benefit many but may feel intrusive and that motivational support should be balanced with patient autonomy is broadly consistent with the Results. However, consider replacing “intrusive” with wording closer to the data (e.g., “unnecessary”/“burdensome”), unless you have verbatim support for “intrusive,” to avoid any perception of over-interpretation.

Lines 568–570: “These forums also enabled reflection and evaluation through progress reviews and barrier discussions, facilitating adaptation, a core mechanism for sustaining implementation in dynamic clinical settings.” The Results support that weekly meetings were structured, agenda-driven, and used for progress monitoring/coordination. However, the manuscript does not currently provide clear empirical examples of adaptations arising from these meetings, nor does it measure sustainability mechanisms. Please consider tightening this statement to what is directly supported and/or add concrete examples of adaptations from the FGDs. If you retain “adaptation” and “sustaining implementation,” frame this more explicitly as interpretive/hypothesis-generating and support it with appropriate implementation literature.

Line 580: The statement that evidence regarding effects on substance use outcomes is “mixed” requires an explicit citation. If the sources already cited in the paragraph support this claim, attach them directly to this sentence; otherwise, please add the most appropriate citations.

Lines 590–593: “An additional and unanticipated outcome was enhanced collaboration across units… improved referral processes across services…” This interpretation is largely supported by the qualitative results. Two refinements are recommended: (1) qualify that these are participant-reported perceptions from Braining team members, not unit-wide staff, and (2) consider narrowing “across services” to “between the two units” unless you have evidence of broader service-level effects. In addition, because this is in the Discussion, it would be strengthened by grounding the observation in relevant organizational/leadership theory and discussing implications.

Lines 594–598: This passage is currently vague. It is unclear what “these findings” refer to, particularly given that the preceding paragraph focuses on cross-unit collaboration. If this section is intended to extend the discussion of perceived work-related benefits of Braining, please strengthen the transition and make explicit that these are perceived benefits reported by Braining team participants. In its current form, the text risks implying measurement of job satisfaction, stress, or workload that was not directly assessed. Also, please verify the citation; I believe the appropriate Quadruple Aim reference is: Bodenheimer T, Sinsky C. From triple to quadruple aim: care of the patient requires care of the provider. Ann Fam Med. 2014;12:573–576. doi:10.1370/afm.1713.

Reviewer #2: (No Response)

7. PLOS authors have the option to publish the peer review history of their article (what does this mean?). If published, this will include your full peer review and any attached files.

Reviewer #1: **Yes:**Temitope Ogundare, MD, MPH

Reviewer #2: **Yes:**Julianah Mosanya

---

## [Author Response · Author response to Decision Letter 2]

5 Apr 2026

Response letter

Reviewer #1

I thank the authors for their efforts and careful revision of the manuscript. The current draft is substantially improved compared with the previous version. However, several issues still need to be addressed.

Response to reviewer 1: We sincerely thank the reviewer for their careful evaluation of our revised manuscript and for their constructive and encouraging feedback. We appreciate the recognition that the manuscript has been substantially improved. We have carefully addressed all remaining comments and revised the manuscript accordingly to further enhance clarity, precision, and transparency. Our detailed responses are provided below, and the changes in the manuscript are marked in yellow.

1. Line 162: “Only Braining team members agreed to participate (n=8).” Please ensure that the reporting of the qualitative results and the Discussion consistently reflect this. Using the term “HCWs” throughout may be misleading regarding who contributed qualitative data. You note this in the Limitations, which is helpful; however, the wording in Results/Discussion should also be consistently aligned.

Response: We thank the reviewer for this important clarification. We have revised the manuscript throughout the Results (line 197) and Discussion (line 467) to consistently distinguish between healthcare workers (HCWs) in general and the specific focus group discussion participants (Braining team members, n=8). As a result, wording has been updated in multiple places and marked in yellow to ensure that the source of each finding is described accurately and consistently.

2. Lines 201–202: “Of the 29 HCWs working at the units, a total of 16 participated in this study. Specifically, 14 HCWs completed the surveys, of whom 8 also participated in the FGDs.” This description is unclear as written. Please clarify the total number of participants.

Response: We thank the reviewer for highlighting this lack of clarity. We have revised the text to explicitly report the total number of participants and the overlap between survey and qualitative data. The revised description now clarifies how many participants contributed to each component and how many participated in both, to ensure that the sample is transparent and easy to interpret.

Line 201-204: Of the 29 HCWs working at the units, 16 participated in the study: 14 completed the surveys, eight participated in the FGDs, and six contributed to both. Consequently, eight participants provided survey data only, two provided qualitative data only, and six contributed to both components.

3. Lines 551–553: The claim that check-ins benefit many but may feel intrusive and that motivational support should be balanced with patient autonomy is broadly consistent with the Results. However, consider replacing “intrusive” with wording closer to the data (e.g., “unnecessary”/“burdensome”), unless you have verbatim support for “intrusive,” to avoid any perception of over-interpretation.

Response: We agree and have revised the wording to better reflect the empirical data. The term “intrusive” has been replaced with “unnecessary” to avoid potential over-interpretation.

Line 552-554: Although regular check-ins were beneficial for many patients, others found them unnecessary, again underscoring the importance of assessing needs to balance motivational support with patient autonomy.

4. Lines 568–570: “These forums also enabled reflection and evaluation through progress reviews and barrier discussions, facilitating adaptation, a core mechanism for sustaining implementation in dynamic clinical settings.” The Results support that weekly meetings were structured, agenda-driven, and used for progress monitoring/coordination. However, the manuscript does not currently provide clear empirical examples of adaptations arising from these meetings, nor does it measure sustainability mechanisms. Please consider tightening this statement to what is directly supported and/or add concrete examples of adaptations from the FGDs. If you retain “adaptation” and “sustaining implementation,” frame this more explicitly as interpretive/hypothesis-generating and support it with appropriate implementation literature.

Response: We thank the reviewer for this important observation. We have revised this section to more closely reflect the empirical data and to avoid overinterpretation. The description of the meetings has been adjusted to align with the Results, emphasising their structured and coordinating role. Where reference to adaptation is retained, it is now more cautiously framed as an interpretive perspective (e.g., “appeared to” / “may have supported”), rather than a directly measured mechanism. We have also clarified the link to implementation literature regarding adaptation and sustaining implementation.

Lines 567–573: The structure of regular meetings appeared to support the team in assessing needs (patient-level and organisational challenges) and context (local workflows, resources, unit priorities), planning (coordinated recruitment, delivery, and barrier management), and tailoring strategies (procedure adjustments based on insights). These forums also enabled reflection and evaluation through progress reviews and discussions of barriers, which may have supported ongoing adaptation processes. Such processes are considered important for sustaining implementation in dynamic clinical settings (35).

5. Line 580: The statement that evidence regarding effects on substance use outcomes is “mixed” requires an explicit citation. If the sources already cited in the paragraph support this claim, attach them directly to this sentence; otherwise, please add the most appropriate citations.

Response: We thank the reviewer for this important comment. We have revised the sentence to more precisely reflect the evidence base and to ensure that the claim is directly supported by appropriate citations. Specifically, we have clarified that while exercise has been associated with improvements in craving and withdrawal symptoms, effects on substance use outcomes are heterogeneous, and we have aligned the cited references to directly correspond to these distinct aspects of the evidence.

Lines: 581-585: Importantly, although PE has been associated with improvements in craving and withdrawal symptoms (37,48), effects on alcohol and other SUD outcomes are heterogeneous across studies, with some trials and meta-analyses reporting limited or no impact on consumption or binge drinking (49,50).

6. Lines 590–593: “An additional and unanticipated outcome was enhanced collaboration across units… improved referral processes across services…” This interpretation is largely supported by the qualitative results. Two refinements are recommended: (1) qualify that these are participant-reported perceptions from Braining team members, not unit-wide staff, and (2) consider narrowing “across services” to “between the two units” unless you have evidence of broader service-level effects. In addition, because this is in the Discussion, it would be strengthened by grounding the observation in relevant organizational/leadership theory and discussing implications.

Response: We thank the reviewer for this helpful comment. We have revised this section to clarify that these findings reflect participant-reported perceptions from Braining team members, rather than unit-wide staff. We have also narrowed the wording from “across services” to “between the two units,” as suggested. In addition, we have strengthened the discussion by briefly grounding these observations in organisational and implementation perspectives, highlighting the potential role of collaboration and coordination in supporting implementation.

Lines 597-602: An additional and unanticipated outcome, as reported by participants, was enhanced collaboration between the two units. According to participants, Braining strengthened peer-to-peer relationships and improved referral processes, suggesting that innovation may confer organisational benefits beyond patient-level effects. Such improvements in collaboration and coordination are consistent with implementation literature emphasising the role of organisational context and interprofessional collaboration in successful implementation (35).

7. Lines 594–598: This passage is currently vague. It is unclear what “these findings” refer to, particularly given that the preceding paragraph focuses on cross-unit collaboration. If this section is intended to extend the discussion of perceived work-related benefits of Braining, please strengthen the transition and make explicit that these are perceived benefits reported by Braining team participants. In its current form, the text risks implying measurement of job satisfaction, stress, or workload that was not directly assessed. Also, please verify the citation; I believe the appropriate Quadruple Aim reference is: Bodenheimer T, Sinsky C. From triple to quadruple aim: care of the patient requires care of the provider. Ann Fam Med. 2014;12:573–576. doi:10.1370/afm.1713.

Response: We thank the reviewer for this important observation. We have revised this section to improve clarity, strengthen the transition, and more clearly distinguish between work-related perceptions and organisational outcomes. Specifically, we now explicitly clarify that these findings refer to participant-reported perceptions from Braining team members, rather than formally measured outcomes. The text has also been restructured to separate staff-related experiences (e.g., sense of purpose and enjoyment at work) from organisational-level observations (e.g., collaboration between the two units), thereby improving coherence. In addition, we have refined the wording to avoid implying measurement of job satisfaction or workload, and we have chosen to remove “stress”. Finally, we have verified and updated the reference to the Quadruple Aim framework (Bodenheimer & Sinsky, 2014).

Lines 591-607:

Moreover, Braining appeared to positively influence the participants’ perceived sense of purpose and enjoyment at work, raising the question of whether the innovation may also affect job satisfaction, work engagement, and perceived value among HCWs. These observations reflect participant-reported perceptions rather than formally measured outcomes. Such outcomes are particularly relevant given evidence that higher staff engagement is associated with reduced turnover and improved workforce productivity in healthcare settings (51).

An additional and unanticipated outcome, as reported by participants, was enhanced collaboration between the two units. According to participants, Braining strengthened peer-to-peer relationships and improved referral processes, suggesting that innovation may confer organisational benefits beyond patient-level effects. Such improvements in collaboration and coordination are consistent with implementation literature emphasising the role of organisational context and interprofessional collaboration in successful implementation (35). Taken together, these participant-reported findings highlight that HCWs may experience distinct, often overlooked outcomes, such as satisfaction and work experience with delivering an innovation. As emphasised within the Quadruple Aim framework, improving the work-life and well-being of HCWs is a critical consideration, as their experiences may influence the sustainability of innovations in real-world practice (52).

---

## [Decision Letter · Decision Letter 2]

13 Apr 2026

Getting psychiatry on the move – Implementation and evaluation of Braining, a structured physical exercise intervention in outpatient psychiatry: a convergent-parallel mixed methods study

PONE-D-25-52478R2

Dear Dr. Mac,

We’re pleased to inform you that your manuscript has been judged scientifically suitable for publication and will be formally accepted for publication once it meets all outstanding technical requirements.

Kind regards,

Anthony A. Olashore, MBCHB, PhD.

Academic Editor

PLOS One

Additional Editor Comments (optional):

Reviewers' comments:

Reviewer's Responses to Questions

**Comments to the Author**

1. If the authors have adequately addressed your comments raised in a previous round of review and you feel that this manuscript is now acceptable for publication, you may indicate that here to bypass the “Comments to the Author” section, enter your conflict of interest statement in the “Confidential to Editor” section, and submit your "Accept" recommendation.

Reviewer #1: All comments have been addressed

2. Is the manuscript technically sound, and do the data support the conclusions?

Reviewer #1: Yes

3. Has the statistical analysis been performed appropriately and rigorously? 

Reviewer #1: Yes

4. Have the authors made all data underlying the findings in their manuscript fully available?

Reviewer #1: Yes

5. Is the manuscript presented in an intelligible fashion and written in standard English?

Reviewer #1: Yes

6. Review Comments to the Author

Reviewer #1: The authors have revised the manuscript and have addressed all the comments. Mansucript is deemed acceptable for publication

7. PLOS authors have the option to publish the peer review history of their article (what does this mean?). If published, this will include your full peer review and any attached files.

Reviewer #1: **Yes:**Temitope Ogundare, MD, MPH

---

## [Editor Report · Acceptance letter]

PONE-D-25-52478R2

PLOS One

Dear Dr. Mac,

I'm pleased to inform you that your manuscript has been deemed suitable for publication in PLOS One. Congratulations! Your manuscript is now being handed over to our production team.

Kind regards,

on behalf of

Dr. Anthony A. Olashore

Academic Editor

PLOS One